# How do smoking, vaping, and nicotine affect people with epilepsy and seizures? A scoping review protocol

Jackson A. Narrett[1], Waleed Khan[2], Melissa C. Funaro[3], Jeremy J. Moeller[1] *

1 Department of Neurology, Yale University School of Medicine, New Haven, Connecticut, United States of America, 2 Yale University School of Medicine, New Haven, Connecticut, United States of America, 3 Harvey Cushing/John Hay Whitney Medical Library, Yale University, New Haven, Connecticut, United States of America

* Jeremy.Moeller@yale.edu

## Abstract

**Data Availability Statement:** No datasets were generated or analysed during the current study. All relevant data from this study will be made available upon study completion.

### Background

Epilepsy is a prevalent disease that requires personalized care to control seizures, reduce side effects, and ameliorate the burden of comorbidities. Smoking is a major cause of preventable death and disease. There is evidence that patients with epilepsy smoke at high rates and that smoking may increase seizure frequency. However, there is a lack systematically synthesized evidence on the interactions between epilepsy and seizures and smoking, tobacco use, vaping, and smoking cessation.

### Methods and analysis

This scoping review protocol guided by the Joanna Briggs Institute Manual for Evidence Synthesis and the PRISMA Extension for Scoping Reviews will investigate what is known about the interactions between smoking and epilepsy. This review will include the population of persons with all types of epilepsy or seizures and examine an inclusive list of concepts including tobacco use, vaping, nicotine replacement, and smoking cessation. The MEDLINE, Embase, APA Psycinfo, CINAHL, Cochrane, Scopus, and Web of Science databases will be searched. Following systematic screening of records, data will be charted, synthesized, and summarized for presentation and publication.

### Ethics and dissemination

No ethical approval is required for this literature-based study. The results of this scoping review will be submitted for publication in a peer-reviewed journal. This synthesis will be informative to clinicians and direct further research that may improve health outcomes for people with epilepsy.

**Funding:** The authors received no specific funding for this work.

**Competing interests:** The authors have declared that no competing interests exist.

## Registration

This protocol is registered with the Open Science Framework (DOI: https://doi.org/10.17605/OSF.IO/D3ZK8).

## Introduction/Rationale

There are an estimated 50 million people living with epilepsy worldwide [1]. People with epilepsy are at risk for reductions in quality of life and well-being not only due to increases in seizure frequency but also due to neuropsychological and medical comorbidities [2, 3]. A comprehensive approach to epilepsy management requires an individualized approach to seizure control, minimization of side effects from medications and surgery, and optimization of comorbidities and lifestyle [3, 4].

Cigarette smoking is a leading cause of preventable death [5]. In the United States, some demographic groups use tobacco at increased rates [6]. There is evidence that people with epilepsy may experience such a disparity [7–9]. Specifically, while rates of smoking declined amongst all adults in the United States between 2010 and 2017, such a decline was not seen in people with epilepsy, with as many as 25% of people with epilepsy in the United States actively smoking in 2017 [9]. Cigarette smoking may be associated with an increased risk of seizures in patients with epilepsy [10]. Furthermore, smoking is known to contribute to various medical comorbidities that people with epilepsy experience increased risk for, such as reduced bone density and cardiovascular disease [11, 12]. Therefore, people with epilepsy who smoke are at risk for both seizure related, and non-seizure related poor health outcomes. Some evidence also suggests that people who smoke may be at an increased risk of developing epilepsy [13].

Nicotine, the primary addictive chemical in tobacco smoke, has been shown to have a proconvulsive effect in animal models. This effect is attributed to the activation of nicotinic acetylcholine receptors (nACHRs) in the brain [14]. Animal models with mutant nACHR subunits with slower desensitization have increased sensitivity to nicotine induced seizures [14]. Likewise, animals with deficiency in subunits that increase the receptors' function have been shown to have decreased sensitivity to nicotine induced seizures. While the precise cellular and network mechanisms that underly nicotine induced seizures are not fully elucidated, the cholinergic nuclei in the brainstem and basal forebrain play key roles in arousal and attention via their effects on cortical tone and modulation of such systems could therefore theoretically affect the seizure threshold. Additionally, nicotine at subconvulsive doses given chronically has been demonstrated to have a seizure kindling effect in animal models [14].

Nicotine may however have anticonvulsant effects in certain contexts. Autosomal Dominant Sleep Related Hypermotor Epilepsy (ADHSE, previously Autosomal Dominant Nocturnal Frontal Lobe Epilepsy) is a rare inherited focal onset epilepsy syndrome associated with mutations in genes that encode the subunits of nACHRs [15]. In vitro expression of such mutant receptors has shown an increased sensitivity to ligands [16]. Interestingly, nicotine may improve seizure control in people with ADSHE due to either receptor desensitization that results from chronic exposure or an alteration of the polymerization of nACHRs to favor stoichiometry with lower ligand affinity [15].

Both the proconvulsant and anticonvulsant effects of nicotine described above have been demonstrated clinically. Seizures have been reported in cases of nicotine poisoning from nicotine patches reported to poison control centers [17]. Seizures have also been reported following exposure to vaped nicotine, or nicotine containing fluids to be used in vaping devices [18–20].

In small clinical studies of patients with ADSHE, nicotine therapy has demonstrated antiseizure effects [21, 22].

The effect of tobacco smoke on the seizure threshold may be due to the effects of nicotine described above, or because of other chemical compounds. For example, tobacco smoke contains chemicals such as arsenic, ammonia, and acetone that have been shown to induce seizures under certain conditions in animal studies [23]. Furthermore, tobacco smoke has been demonstrated to alter the metabolism of various compounds metabolized by the cytochrome P450 and UDP-glucuronyl transferase systems [24, 25]. Affected compounds could include medications or substances that lower the seizure threshold and antiseizure medications. For example, smoking has been shown to reduce serum levels of lamotrigine [26].

Clinicians who care for people with epilepsy and seizures have a unique opportunity to positively impact the health of their patients by employing a holistic approach to neurologic care informed by an understanding of the interactions of epilepsy with lifestyle factors including smoking. Choice of epilepsy directed therapy such as antiseizure medication or surgery should be tailored to an individual patient's comorbidities, risk factors, and preferences. Moreover, smoking cessation pharmacotherapy must be considered carefully in patients with epilepsy as bupropion is contraindicated and varenicline is recommended for cautious use in patients at risk for seizures [27]. Currently, the interactions between smoking, nicotine, or vaping and epilepsy or seizures are not well understood and practicing clinicians lack evidence-based guidance to address these issues. To synthesize the evidence on this topic, define gaps in knowledge, and assess opportunities for further research in this area, we aim to conduct a scoping review.

## Study objectives

The objective of this scoping review is to systematically scope the existing research on the interaction of tobacco and related products and epilepsy. This review is directed towards the following questions:

1. What is the prevalence of tobacco use among people with epilepsy?

2. Does tobacco use/smoking affect seizure control in patients with epilepsy?

3. Does tobacco use, smoking, or vaping (e-cigarette use) increase an individual's risk of developing epilepsy?

4. What is known about vaping and seizures or epilepsy?

5. What is known about smoking cessation in the epilepsy population?

6. Do antiseizure medications interact with nicotine, smoking, or vaping?

We will identify the quantity and heterogeneity of studies investigating the above questions, summarize the results of these studies, and identify gaps in knowledge. This study aims to be informative to practicing clinicians caring for patients with epilepsy and researchers designing future studies to elucidate interactions between epilepsy and tobacco use.

## Methods and analysis

This scoping review protocol is guided by the framework described by the Joanna Briggs Institute (JBI) [28] and the PRISMA Extension for Scoping Reviews (PRISMA-ScR): Checklist and Explanation [29]. The PRISMA-ScR checklist with protocol relevant sections completed is included here as a S1 Checklist. These resources will be referenced throughout study execution

and reporting. This protocol is registered with the Open Science Framework (DOI: https://doi.org/10.17605/OSF.IO/D3ZK8)

## Stage 1: Identifying the question

Research questions were identified through nonsystematic review of literature published on this topic and through discussion between authors. The relevant questions are listed in the study objectives section of this protocol. These questions may be revised, amended, or potentially replaced during an iterative review process.

## Stage 2: Identifying relevant studies

The Population-Concept-Context (PCC) framework is recommended by the Joanna Briggs Institute for scoping reviews and will be used to inform our inclusion criteria [28] (Table 1). We will include peer reviewed articles of all types, book chapters, abstracts, and publications from the grey literature on this topic. Animal studies and non-human laboratory investigations will be excluded. We will include studies published from all years included in the relevant databases. Studies without an available English language article will be excluded. Studies that involve cannabis or cannabinoids will be excluded. We will update relevant concepts through an iterative process during screening.

We have elected to include studies that examine the interaction between smoking and epilepsy or seizures in both adult and pediatric samples. While this could decrease the internal validity of our synthesis, for the purpose of this scoping review, we prioritize capture of extant knowledge in this field. Our preliminary search described in Step 1 of Table 2 demonstrated that significant contributions to our understanding of the interactions of smoking, nicotine, and tobacco with seizures and epilepsy come from the pediatric population. This includes data on toxic exposures in nicotine, vaping and seizures, and experimental studies of nicotine as an antiseizure medication [20, 22, 30]. Our preliminary search also demonstrated that the literature on any specific type of epilepsy or tobacco/nicotine use pattern is limited. To scope what is known about these topics in an inclusive review that may inform further research, we plan to use a comprehensive list of concepts in our inclusion criteria.

**Search strategy.** We will use a three-step search process to identify records to include in this scoping review as recommended by the Joanna Briggs Institute [28] (Table 2).

An experienced medical librarian (MCF) was consulted on review methodology and search technique. A medical subject heading (MeSH) analysis of known key articles provided by the research team [mesh.med.yale.edu] was done and scoping searches were conducted in each database. An iterative process was used to translate and refine the searches. To maximize sensitivity, the formal search will use controlled vocabulary terms and synonymous

**Table 1. The Population-Concept-Context framework for this scoping review protocol.**

| P—Population | Persons of all ages with epilepsy (Including all types) |
|---|---|
| C—Concept | Cigarette smoking, tobacco use, vaping and electronic cigarette use, nicotine replacement, smoking cessation. |
| | Potential outcomes include prevalence of product use, seizure frequency, prevalence of refractory epilepsy, incidence of seizure, incidence or prevalence of epilepsy diagnosis, incidence of SUDEP, cessation rate, serum antiseizure medicine levels. |
| C—Context | Studies that investigate this topic in any clinical or community context will be included. Records from any country will be included in our review. Records must have an available English language article to be included for data extraction. |

**Table 2. The three-step search process to be used in this scoping review.**

| | |
|---|---|
| Step 1 | A preliminary search for background information and previously conducted systematic or scoping reviews related to this topic was conducted prior to protocol design by searching PubMed. |
| Step 2 | Searches will be conducted using our final identified search string in the following databases: MEDLINE, Embase, APA Psycinfo, CINAHL, Cochrane, Scopus, and Web of Science. No date or language limits will be imposed on the search. |
| | The following search terms will be included: |
| | Smoking |
| | Electronic cigarette |
| | Vaping |
| | Nicotine |
| | Tobacco |
| | Epilepsy |
| | Seizure |
| | Convulsions |
| | Anticonvulsive |
| | Antiseizure |
| | Antiepileptic |
| Step 3 | We will identify additional studies using a focused and systematic search of cited studies in articles selected for inclusion using the citationchaser software program [https://estech.shinyapps.io/citationchaser/]. |
| | We will limit inclusion of the grey literature to conference abstracts and registered clinical trials. |

free-text words to capture the concepts of "tobacco smoking" and "epilepsy". We will include search terms to capture studies on specific tobacco products utilized in various geographical settings to be as inclusive as possible. The search strategy was peer reviewed by a second librarian, not otherwise associated with the project, using the PRESS standard [31]. An example OVID MEDLINE search strategy is included in Fig 1. During the data extraction process, reviewers will check for additional relevant cited and citing articles using included studies. To capture recently published articles, a second database search will be rerun before publishing the paper.

## Stage 3: Study selection

Search results will be pooled in EndNote 20 and de-duplicated [www.endnote.com]. This set will be uploaded to Covidence [www.covidence.org] for screening. Three authors (JAN, WK, JJM) will independently screen titles and abstracts and subsequently full text records using the prespecified inclusion criteria outlined in Table 1 of this protocol. Conflicting decisions from both title and abstract and full text screening processes will be resolved by unanimous consensus of the three screening authors.

## Stage 4: Charting the data

Three authors (JAN, WK, JJM) will participate in the data extraction process. A shared form will be generated for the purposes of data extraction. This form will include the following data fields: year of publication, study design, sample size, intervention or comparator, primary outcome measure, secondary outcome measure, primary result, secondary result. This form will be updated by the authors through an iterative process to ensure capture of relevant data. Additional fields will be added to facilitate capturing the scope of the published data. Following publication of our results, raw data tables will either be uploaded with the publication or maintained by the corresponding author to be available upon request.

| | |
|---|---|
| 1 | exp smoking/ or vaping/ or smoking cessation/ or smoking reduction/ or "tobacco use cessation"/ or (beedi* or bidi* or cigar* or cigarette* or cigarillo* or dhumti or dokha or ecig* or e-cig* or "electronic cigar*" or gutka or imqmik or khaini or kretek* or makla or maassel or "mu'assel" or naswar or nicotine or tobacco or perique or shisha or smoking or snuff or snus or "thoc lao" or tobacco or vape or vaping).tw,kf. |
| 2 | Anticonvulsants/ or exp Epilepsy/dt or exp Seizures/dt or ((anticonvuls* or antiseizure or antiepileptic) adj2 (medication* or drug* or agent*)).tw,kf. |
| 3 | exp Epilepsy/ or exp Seizures/ or (epilep* or seizure* or convuls* or anticonvulsant* or antiepileptic*).tw,kf. |
| 4 | 2 or 3 |
| 5 | 1 and 4 |
| 6 | 5 not (Animals/ not (Animals/ and Humans/)) |

**Fig 1. Example OVID MEDLINE search strategy for this scoping review.**

## Stage 5: Collating, summarizing, and reporting the results

The PRISMA-ScR extension will inform the reporting and discussion of the results of this review [28]. A PRISMA diagram will be generated to summarize the search, screening, and study identification procedure [29].

The data will be summarized in a tabular format to clearly present the relevant details of the studies reviewed. A bubble plot will be used to depict the number of studies published per year in various categories including but not limited to epidemiology, seizure risk, cigarettes, vaping, nicotine replacement, smoking cessation, and medication–tobacco/nicotine interaction. Additional tables and figures will be added to clearly present the results of this review.

A detailed text synthesis of the findings of the included studies will be composed as the results section of a manuscript describing this scoping review. This synthesis will be guided by the research questions included in the study objectives section of this protocol.

Critical appraisal of study quality will be conducted with tools recommended by the Joanna Briggs Institute [32]. The relevant JBI checklist will be completed for each included study. A synthesis of the critical appraisal of included studies will be described in the results section of our manuscript.

Our manuscript will include a discussion section that will summarize the results of our review, highlight important and or surprising results, discuss additional research questions identified, describe the limitations of this review, and comment on the quality of the evidence identified.

### Stage 6: Consultation

Prior to reporting the results of this scoping review of the literature we will share our manuscript, tables, and materials with expert epileptologists. We will collect feedback to improve our reporting and summary of the results of our scoping review. We do not plan to conduct community or focus group consultation prior to publishing the results of this scoping review.

**Ethics and dissemination.**   No ethical approval was required for this study.

The synthesized results of this scoping review will be detailed in a manuscript and submitted for publication in a peer-reviewed journal. Conference abstracts may be submitted detailing the procedures described in this report and the results of this study prior to submission for publication.

## Discussion

This scoping review will identify the scope of what is known about the interactions between smoking and epilepsy. A strength of our protocol is an inclusive search strategy to capture the spectrum of this topic by including various forms of tobacco use, vaping, nicotine replacement, and smoking cessation. A limitation of this study is the choice to limit the review to human studies only. This will exclude data from animal studies on the science of tobacco and epilepsy. We believe that this choice is appropriate to generate a clinically relevant review. Our scoping review aims to be useful to clinicians who care for patients with epilepsy and to inform the design of further research studies to investigate this topic. An improved understanding of the interaction of smoking and epilepsy may lead to better outcomes for patients with epilepsy who experience the comorbidity of tobacco use through informed personalized epilepsy treatment and smoking cessations strategies.

## Supporting information

**S1 Checklist. Preferred Reporting Items for Systematic reviews and Meta-Analyses extension for Scoping Reviews (PRISMA-ScR) checklist.**
(DOCX)

## Author Contributions

**Conceptualization:** Jackson A. Narrett, Melissa C. Funaro, Jeremy J. Moeller.

**Methodology:** Jackson A. Narrett, Waleed Khan, Melissa C. Funaro, Jeremy J. Moeller.

**Writing – original draft:** Jackson A. Narrett.

**Writing – review & editing:** Jackson A. Narrett, Waleed Khan, Melissa C. Funaro, Jeremy J. Moeller.

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
