## [Decision Letter · Decision Letter 0]

14 Mar 2023

PONE-D-22-27041What is known about Tobacco Use, Smoking, Vaping, and Epilepsy? A Scoping Review ProtocolPLOS ONE

Dear Dr. Narrett,

Thank you for submitting your manuscript to PLOS ONE. After careful consideration, we feel that it has merit but does not fully meet PLOS ONE’s publication criteria as it currently stands. Therefore, we invite you to submit a revised version of the manuscript that addresses the points raised during the review process.

We look forward to receiving your revised manuscript.

Kind regards,

Sathish Rajaa

Academic Editor

PLOS ONE

Journal Requirements:

Additional Editor Comments (if provided):

The authors attempts on generating evidence on the pattern smoking, vaping and other tobacco use among patients with seizures is appreciated, however there are a few clarifications that are needed from our side:

General concerns:

1. The author need to revisit the writeup, and consider expanding the abbreviations

Title:

1. The title appears very vague, the objectives and the research question is not reflected in the topic. Consider providing a more specific topic

Introduction:

1. The introduction is very non specific, the biochemical and social explanation for studying the effect of smoking, vaping and other tobacco use among patients with seizures needs to be elaborated.

2. Consider adding more literature that reports a biochemical explanation for studying the pattern and prevalence of tobacco and other related products among seizure patients

Methods:

The scoping review confines to the PRISMA with ScR extension,

However we need clarification on the following:

1. Please provide more clarification on the study population, why to include children? when the objective is to see the pattern of tobacco and smoking?

2. Elaborate more on the eligibility criteria (e.g., years considered, language, and publication status), and provide a rationale

3. How are you planning to handle the huge grey literature?

4. Provide a still more comprehensive search strategy.

5. Why no efforts are made to access the quality of included studies using a more accepted tool like the CASP checklist?

6. Including all types of epilepsies and investigating on all patterns of tobacco use including all possible literature sources? kindly provide a rational for including all epilepsy types and tobacco use patterns? why not focus on a specific type?

7. Expand 'Tobacco products' in the search strategy thereby the authors don't miss out on any of the types that are available?

8. How are you planning to discuss the results? are any frameworks are planned to be used? if so do mention the framework that is planned to be adopted

Reviewers' comments:

Reviewer's Responses to Questions

**Comments to the Author**

1. Does the manuscript provide a valid rationale for the proposed study, with clearly identified and justified research questions?

Reviewer #1: Yes

Reviewer #2: Yes

2. Is the protocol technically sound and planned in a manner that will lead to a meaningful outcome and allow testing the stated hypotheses?

Reviewer #1: Yes

Reviewer #2: Yes

3. Is the methodology feasible and described in sufficient detail to allow the work to be replicable?

Reviewer #1: Yes

Reviewer #2: Yes

4. Have the authors described where all data underlying the findings will be made available when the study is complete?

Reviewer #1: Yes

Reviewer #2: Yes

5. Is the manuscript presented in an intelligible fashion and written in standard English?

Reviewer #1: Yes

Reviewer #2: Yes

6. Review Comments to the Author

You may also provide optional suggestions and comments to authors that they might find helpful in planning their study.

Reviewer #1: I do value the importance of smoking in epilepsy pathogenesis and pharmacotherapy and would like the authors to add another question: the impact of passive smoking during pregnancy on thr incidence of epilepsy development in offsprings via retrospective evaluation.

Reviewer #2: The scoping review protocol was prepared according to the standardized method stipulated by JBIM. The review objectives are all clear and well defined. Just one suggestion, if the keyword for Tobacco should expand: 'Tobacco products' to ensure that the authors won't miss any kind of specific tobacco products that used in other studies from local settings such as kretek, Shisha, Bidis. The rest is all good and clearly written.

7. PLOS authors have the option to publish the peer review history of their article (what does this mean?). If published, this will include your full peer review and any attached files.

Reviewer #1: **Yes: **Mohamed Mostafa

Reviewer #2: No

---

## [Author Response · Author response to Decision Letter 0]

11 May 2023

1. The title appears very vague, the objectives and the research question is not reflected in the topic. Consider providing a more specific topic

We thank the editor for this helpful comment. We have rewritten our title to more specifically describe our topic of investigation.

We have changed our title to: "How do smoking, vaping, and nicotine affect people with epilepsy and seizures? A scoping review protocol"

1. The introduction is very non specific, the biochemical and social explanation for studying the effect of smoking, vaping and other tobacco use among patients with seizures needs to be elaborated.

We thank the editor for this helpful comment. We agree that our manuscript could benefit from further explanation of our study's rationale.

We have added content to the introduction to describe the biochemical rationale for studying the effects of smoking, vaping, and tobacco use among patients with epilepsy or seizures. To the second paragraph of the introduction, we have added two sentences. One stating the relevance of our topic to the health outcomes of people with epilepsy. The other comments on evidence suggesting that people who smoke may be at risk for developing epilepsy. We have added three paragraphs to the introduction section that describe the biochemical basis for the proconvulsive effects of nicotine, a possible role of nicotine as an anticonvulsant, and potential effects of the unique exposure of tobacco smoke on the seizure threshold. We have also modified a paragraph that discusses the clinical relevance of the basic science findings described in our introduction. We have made some additional small edits to wording and phrasing in the introduction to clarify the aims of our study.

2. Consider adding more literature that reports a biochemical explanation for studying the pattern and prevalence of tobacco and other related products among seizure patients

We thank the editor for this helpful comment. Our preliminary searches on this topic did not reveal studies reporting data on a biochemical cause of increased smoking rates in people with epilepsy or seizure. Our rationale for studying this topic is related to the epidemiologic data suggesting that people with epilepsy may smoke with increased rates. This could be hypothesized to be either due to effects of having epilepsy such as increased rates of comorbid mental illness, social factors such as lower socioeconomic status, or due a shared diathesis for addictive behavior and seizures due to the behavior of pathological neural networks in the brain. We believe that this is an open question and we hope that our systematic approach to scoping the relevant literature may reveal further studies that comment on this question. For now, we will further describe the epidemiologic finding that serves as our rationale for our study. If we find additional studies supporting increased rates of smoking in people with epilepsy, particularly from different geographical reasons, we will describe this trend and any relevant data on the hypotheses mentioned above in the results and discussion sections of the manuscript that will report this study.

We have added two sentences to the second paragraph of the introduction/rationale section describing tends of smoking rates amongst people with epilepsy in the United States.

1. Please provide more clarification on the study population, why to include children? when the objective is to see the pattern of tobacco and smoking?

We thank the editor for this helpful comment. Our aim is to be as inclusive as possible to systematically scope the published literature on smoking, vaping, tobacco, nicotine use, and epilepsy or seizures. We do appreciate that inclusion of both adult and pediatric patients introduces a heterogeneity to our study. However, we believe that including pediatric studies is additive. Prelimary search identified additive studies pediatric population, including data on seizures provoked by nicotine and on treatment of ADSHE. We believe that conducting a broad scoping review will be most useful to framing what is known on our topic of interest and informing further research.

We have added a paragraph describing our rationale for including both adult and pediatric patients to our section "Stage 2: Identifying relevant studies"

2. Elaborate more on the eligibility criteria (e.g., years considered, language, and publication status), and provide a rationale

We will include studies from all years available in the individual databases (this might be different for each one). We will include English Language studies. Our first paragraph of the stage 2 section described publication status to be included. 

We have added "Studies without an available English language article will be excluded." To the text of our identifying relevant studies section. "Records must have an available English language article to be included for data extraction." is also included in the "Context" section of our Population-Concept-Context Table (table 1). We have added a sentence to the first paragraph of the stage 2 section stating that we will include studies from all years included in relevant databases.

3. How are you planning to handle the huge grey literature?

Thank you for this helpful comment. We have revised our plan for the grey literature. We will limit inclusion of grey literature to the conference abstracts and clinical trial protocols identified by our search strategy. We believe that this approach is reasonable and that our inclusive scoping review of the academic literature will identify what is known on our topic of interest.

We have added a sentence to step 3 of table two stating how we will limit the our inclusion of the grey literature on this topic.

4. Provide a still more comprehensive search strategy.

Thank you for this comment. We have included a three step description of our search strategy in table 2. We describe our formal search of the literature in the search strategy section of our paper. We include an OVID MEDLINE example search strategy. We believe that our search could be replicated by a medical librarian with appropriate training and experience. In order improve our communication of the search strategy, we will move the MEDLINE search strategy from Appendix 1 to Figure 1 in the body of the manuscript.

We have moved the MEDLINE search strategy from Appendix 1 to Figure 1.

5. Why no efforts are made to access the quality of included studies using a more accepted tool like the CASP checklist?

Thank you for this helpful comment. Our original plan was to use the JBI checklists only if 5 or more studies of a single type were identified. We recognize that our review will be improved by systematic critical appraisal and we will now plan to use the JBI checklists for every article identified. In making this decision we reviewed both of CASP checklists and JBI checklists. While both are excellent tools, we are opting to use the JBI checklists as they include only multiple choice fields without free text. This will improve our ability to synthesize this data in a table or figure.

We have edited a sentence in the third paragraph of our Stage 5 section of the manuscript describing our plan.

6. Including all types of epilepsies and investigating on all patterns of tobacco use including all possible literature sources? kindly provide a rational for including all epilepsy types and tobacco use patterns? why not focus on a specific type?

Thank you for this helpful comment. Our review aims to systematically scope what is known about the effects of tobacco use, vaping, and nicotine related to seizures and epilepsy. Our preliminary search, described in step 1 of our search process table (table 2), demonstrated that the literature on this topic is highly heterogeneous and data on any given type of epilepsy or tobacco use pattern would be limited. By conducting an inclusive review, we believe we can better define the landscape of what is known on these topics. This scoping review may identify areas that could benefit from subsequent systematic review and/or metanalysis

We have added a sentence to the second paragraph of the Stage 2 section of our manuscript describing our rationale for including all epilepsy types and tobacco use patterns.

7. Expand 'Tobacco products' in the search strategy thereby the authors don't miss out on any of the types that are available?

Thank you for this helpful comment. We agree that this would improve our search strategy.

We have added a sentence to the second paragraph of our search strategy section describing this change. We have updated Figure 1 to include these terms in our search.

8. How are you planning to discuss the results? are any frameworks are planned to be used? if so do mention the framework that is planned to be adopted

Thank you for this helpful comment. Our review will include a discussion section. We will discuss the result of our review within the context of related research in the field of epilepsy. We will highlight important or surprising findings from our results. We will discuss additional research questions identified by our review. We will also discuss the limitations of our review and comment on the quality of evidence identified. This discussion will be informed by the PRISMA-ScR checklist. 

We have added to the text of the Stage 5 section "Our manuscript will include a discussion section that will highlight important and or surprising results from this review, discuss additional research questions identified, discuss the limitations of this review, and comment on the quality of the evidence identified." We have also added "and discussion" to the first sentence of the Stage 5 section of our manuscript to clarify that the PRISMA- ScR checklist will be used to inform the writing of our discussion section in addition to our results section.

I do value the importance of smoking in epilepsy pathogenesis and pharmacotherapy and would like the authors to add another question: the impact of passive smoking during pregnancy on thr incidence of epilepsy development in offsprings via retrospective evaluation.

We thank the reviewer for this input. We agree that the neurologic outcomes of children of mothers who are exposed to passive smoking during pregnancy is an important topic that could benefit from further research. However, we have chosen not to include the effects of smoking on epilepsy outcomes in children of mothers who smoke in this review. Our review is quite broad and seeks to scope a body of literature to address multiple questions. We are concerned that adding this as an additional question would limit our ability to synthesize evidence and discuss our results in a single manuscript.

The scoping review protocol was prepared according to the standardized method stipulated by JBIM. The review objectives are all clear and well defined. Just one suggestion, if the keyword for Tobacco should expand: 'Tobacco products' to ensure that the authors won't miss any kind of specific tobacco products that used in other studies from local settings such as kretek, Shisha, Bidis. The rest is all good and clearly written.

Thank you for this helpful comment. We agree that this would improve our search strategy.

We have made the changes described above. (We have added a sentence to the second paragraph of our search strategy section describing this change. We have updated Figure 1 to include these terms in our search.)

---

## [Decision Letter · Decision Letter 1]

20 Jun 2023

How do smoking, vaping, and nicotine affect people with epilepsy and seizures? A scoping review protocol

PONE-D-22-27041R1

Dear Dr. Narrett,

We’re pleased to inform you that your manuscript has been judged scientifically suitable for publication and will be formally accepted for publication once it meets all outstanding technical requirements.

Kind regards,

Sathish Rajaa

Academic Editor

PLOS ONE

Additional Editor Comments (optional):

Dear [Author's Name],

I hope this email finds you well. I am writing to inform you about the status of your manuscript, titled "How do smoking, vaping, and nicotine affect people with epilepsy and seizures? A scoping review protocol", which you submitted to our esteemed journal Plos one. As an Academic Editor, I am pleased to inform you that your manuscript has successfully completed the revision process and has now entered the peer review stage.

I want to express my appreciation for the thorough revisions you have made in response to the reviewers' comments. After carefully evaluating the revised manuscript, I am pleased to inform you that the reviewers have expressed their satisfaction with the changes made. They have acknowledged the improvements and believe that your manuscript is now ready for publication in its current form.

Please note that the acceptance is conditional upon successfully completing the remaining editorial and production processes. Our editorial team will now work diligently to move the manuscript forward, ensuring that all necessary steps are taken to prepare it for publication.

Once again, I want to congratulate you on the successful revisions and the acceptance of your manuscript. I appreciate your dedication and hard work throughout the peer review process. Your research will undoubtedly enrich the scientific discourse within your field.

Thank you for choosing Plos one as the outlet for your important research.

Best regards,

Sathish Rajaa

Reviewers' comments:

Reviewer's Responses to Questions

**Comments to the Author**

1. Does the manuscript provide a valid rationale for the proposed study, with clearly identified and justified research questions?

Reviewer #1: Yes

2. Is the protocol technically sound and planned in a manner that will lead to a meaningful outcome and allow testing the stated hypotheses?

Reviewer #1: Yes

3. Is the methodology feasible and described in sufficient detail to allow the work to be replicable?

Reviewer #1: Yes

4. Have the authors described where all data underlying the findings will be made available when the study is complete?

Reviewer #1: Yes

5. Is the manuscript presented in an intelligible fashion and written in standard English?

Reviewer #1: Yes

6. Review Comments to the Author

You may also provide optional suggestions and comments to authors that they might find helpful in planning their study.

Reviewer #1: I value your addressing of my feedback comment and look forward to getting it published. Thank you for the chance.

7. PLOS authors have the option to publish the peer review history of their article (what does this mean?). If published, this will include your full peer review and any attached files.

Reviewer #1: **Yes: **Mohamed Mostafa

---

## [Editor Report · Acceptance letter]

26 Jun 2023

PONE-D-22-27041R1 

How do smoking, vaping, and nicotine affect people with epilepsy and seizures? A scoping review protocol 

Dear Dr. Narrett:

I'm pleased to inform you that your manuscript has been deemed suitable for publication in PLOS ONE. Congratulations! Your manuscript is now with our production department. 

Kind regards, 

on behalf of

Dr. Sathish Rajaa 

Academic Editor

PLOS ONE